

# Ultrasound image denoising using generative adversarial networks with residual dense connectivity and weighted joint loss

Lun Zhang[1,2] and Junhua Zhang[1]

[1] School of Information Science and Engineering, Yunnan University, Kunming, Yunnan, China
[2] Yunnan Vocational Institute of Energy Technology, Qujing, Yunnan, China

Corresponding author
Junhua Zhang, jhzhang@ynu.edu.cn

## ABSTRACT

**Background**. Ultrasound imaging has been recognized as a powerful tool in clinical diagnosis. Nonetheless, the presence of speckle noise degrades the signal-to-noise of ultrasound images. Various denoising algorithms cannot fully reduce speckle noise and retain image features well for ultrasound imaging. The application of deep learning in ultrasound image denoising has attracted more and more attention in recent years.

**Methods**. In the article, we propose a generative adversarial network with residual dense connectivity and weighted joint loss (GAN-RW) to avoid the limitations of traditional image denoising algorithms and surpass the most advanced performance of ultrasound image denoising. The denoising network is based on U-Net architecture which includes four encoder and four decoder modules. Each of the encoder and decoder modules is replaced with residual dense connectivity and BN to remove speckle noise. The discriminator network applies a series of convolutional layers to identify differences between the translated images and the desired modality. In the training processes, we introduce a joint loss function consisting of a weighted sum of the L1 loss function, binary cross-entropy with a logit loss function and perceptual loss function.

**Results**. We split the experiments into two parts. First, experiments were performed on Berkeley segmentation (BSD68) datasets corrupted by a simulated speckle. Compared with the eight existing denoising algorithms, the GAN-RW achieved the most advanced despeckling performance in terms of the peak signal-to-noise ratio (PSNR), structural similarity (SSIM), and subjective visual effect. When the noise level was 15, the average value of the GAN-RW increased by approximately 3.58% and 1.23% for PSNR and SSIM, respectively. When the noise level was 25, the average value of the GAN-RW increased by approximately 3.08% and 1.84% for PSNR and SSIM, respectively. When the noise level was 50, the average value of the GAN-RW increased by approximately 1.32% and 1.98% for PSNR and SSIM, respectively. Secondly, experiments were performed on the ultrasound images of lymph nodes, the foetal head, and the brachial plexus. The proposed method shows higher subjective visual effect when verifying on the ultrasound images. In the end, through statistical analysis, the GAN-RW achieved the highest mean rank in the Friedman test.

## INTRODUCTION

Ultrasound has been widely used in clinical diagnosis. Compared with CT and MRI, it has the advantages of cost-effectiveness and non-ionizing radiation. However, due to the coherent nature, speckle noise is inherent in ultrasound images (*Singh, Mukundan & Ryke, 2017*). The speckle noise is the primary cause of low contrast resolution and the signal-to-noise ratio. It makes image processing and analysis more challenging, such as image classification and segmentation. Therefore, eliminating speckle noise is of great significance for improving the ultrasound images signal-to-noise ratio and diagnosing disease accurately.

There are various traditional methods for image denoising, which include frequency domain, time-domain and joint time-domain/frequency-domain methods. Among the traditional methods, the most widely used denoising method is based on wavelets (*Jaiswal, Upadhyay & Somkuwar, 2014*; *Srivastava, Anderson & Freed, 2016*; *Gupta, Chauhan & Sexana, 2004*). *Shih, Liao & Lu (2003)* proposed an iterated two-band filtering method to solve the selective image smoothing problem. *Yue et al., (2006)* introduced a novel nonlinear multiscale wavelet diffusion for speckle noise removal and edge enhancement, which proved that this method is better than wavelet-transform alone in removing speckle noise. Among the above methods, speckle noise is transformed into additive noise and removed. Because speckle noise is not purely multiplicative noise, the selection of wavelets is based on experience, which creates artefacts. Traditional methods based on the spatial domain include the Kuan filter (*Kuan et al., 1987*), speckle reducing anisotropic diffusion filter (*Yu & Acton, 2002*) and Frost filter (*Frost et al., 1982*). These methods mainly use local pixel comparison. The nonlocal means (NLM) method was proposed which is based on a nonlocal averaging of all pixels in the image (*Buades, Coll & Morel, 2005*). However, the NLM filter cannot preserve the fine details and edge features in the image. *Dabov et al. (2007)* proposed a block-matching and 3D transform-domain collaborative filtering (BM3D) method, which reduced the computing time and effectively suppressed noise by grouping 3D data arrays of similar 2D image fragments. However, the disadvantage of these methods is that they cannot maintain a balance between noise suppression and image detail preservation.

The development of deep learning provides a perfectly feasible solution for image denoising. *Zhang et al. (2017)* introduced feed-forward denoising convolutional neural network (DnCNN), where residual learning (*He et al., 2016*) was adopted to separate noise from noisy image and batch normalization (BN) (*Salimans & Kingma, 2016*) was integrated to speed up the training process and boost the denoising performance. Using the small medical image datasets, *Jifara et al. (2019)* designed a denoising convolutional neural network with residual learning and BN for medical image denoising (DnCNN-Enhanced). More specifically, they used residual learning by multiplying a very small constant and added it to better approximate the residual to improve performance. *Tian, Xu & Zuo (2020)* proposed a novel algorithm called a batch-renormalization denoising network (BRDNet) for image denoising. This network combines two networks to expand the width to capture more feature information. Meanwhile, BRDNet adopted BN to address small

mini-batch problems and dilated convolution to enlarge the receptive field to extract more feature information. In addition to feed-forward denoising algorithm, there are some algorithms based on the encoder–decoder network. The U-Net is the most widely used encoder–decoder network which used the segmentation of biomedical images (*Ronneberger, Fischer & Brox, 2015*). These are various algorithms based on U-Net for image processing, such as U-Net++ (*Zhou et al., 2019*), residual-dilated-attention-gate network (RDAU-Net) (*Zhuang et al., 2019*), Wasserstein GAN algorithm (RDA-UNET-GAN) (*Negi et al., 2020*), Attention Gate-Dense Network-Improved Dilation Convolution-U-Net (ADID-UNET) (*Raj et al., 2021*), VGG-UNet (*Fawakherji et al., 2019*), Ens4B-UNet (*Abedalla et al., 2021*) and so on. *Park, Yu & Jeong (2019)* designed a densely connected hierarchical image denoising network (DHDN) for removing additive white Gaussian noise of natural images. Based on the U-Net, it applied the hierarchical architecture of the encoder–decoder module with dense connectivity and residual learning to solve the vanishing-gradient problem. *Guo et al. (2019)* suggested training a convolutional blind denoising network (CBDNet) using noisy-clean image pairs and realistic noise model. To further provide an interactive strategy to conveniently correct the denoising results, the noise estimation subnetwork with asymmetric learning was embedded in CBDNet to suppress the underestimation of the noise level. *Couturier, Perrot & Salomon (2018)* applied the deep encoder–decoder network (EDNet) to address additive white Gaussian and multiplicative speckle noises. The encoder module used to extract features and remove the noise, whereas the decoder module recovered a clean image. To yield a performance improvement, there are some methods using generative adversarial network (GAN) in the training phase. *Lsaiari et al. (2019)* performed image denoising using generative adversarial network (GAN). *Yang et al. (2018)* introduced a new CT image denoising method based on GAN with Wasserstein distance and perceptual similarity. *Dong et al. (2020)* developed a custom GAN to denoise optical coherence tomography. *Lee et al. (2020)* proposed a model consisting of multiple U-Nets (MuNet) for three-dimensional neural image denoising. It consisted of multiple U-Nets and using GAN in the training phase. These methods perform well in removing Gaussian noise, but they cannot accurately suppress speckle noise. *Wang, Zhang & Patel (2017)* proposed a set of convolutional layers along with a componentwise division residual layer and a rectified linear unit (ReLU) activation function and BN to remove speckle noise. However, such a method cannot deal with the speckle noise of ultrasound images well.

In this thesis, we proposed a generative adversarial network with residual dense connectivity and weighted joint loss (GAN-RW) to overcome the limitations of traditional image denoising methods and surpass the most advanced performance of ultrasound image denoising. The proposed network consists of a denoising network and a discriminator network. The denoising network is based on U-Net architecture which includes four encoder and four decoder modules. Each block of the encoder and decoder is replaced with residual dense connectivity and BN to remove speckle noise. The discriminator network applies a series of convolutional layers to identify differences between the translated images and the desired modality. In the training processes, we introduced a joint loss function consisting of a weighted sum of the L1 loss, the perceptual loss function and the binary cross-entropy with logit loss (BCEWithLogitsLoss) function. Experiments on natural

images and ultrasound images illustrate that the proposed algorithm surpasses the deep learning-based algorithms and conventional denoising algorithms.

The rest of the paper is organized as follows: Section 2 provides the proposed method and implementation details. Extensive experiments are conducted to evaluate our proposed methods in Section 3. We discuss these results in Section 4 and conclude in Section 5.

## MATERIALS & METHODS

An overview of the proposed network framework for ultrasound image denoising is shown in Fig. 1. In this section, the network architecture is introduced in detail.

### Speckle noise model

Due to speckle noise, ultrasound image processing is a very challenging task. Speckle noise is an interference mode generated by the coherent accumulation of random scattering in the ultrasonic beam resolution element, so it has the characteristics of asymmetrical intensity distribution and significant spatial correlation (*Slabaugh et al., 2006*). This characteristic has an adverse effect on the image quality and interpretability. Because these characteristics are difficult to model, many methods of ultrasound image processing only assume that speckle noise is Gaussian noise, resulting in these speckle noise models are more suitable for X-ray and MRI image than ultrasound image. The gamma distribution (*Sarti et al., 2005*) and Fisher-Tippett distribution (*Michailovich & Adam, 2003*) have been proposed to approximate speckle noise. *Slabaugh et al. (2006)* argued that Fisher-Tippett distribution was suitable for fully formed speckle noise in the ultrasound image. In this article, the speckle noise model of the ultrasound image is given as:

$$v(x,y) = u(x,y) + u(x,y)^r \theta(x,y) \tag{1}$$

where $v(x, y)$ is the pixel location of the speckle noise image, $u(x, y)$ is the pixel location of the noise-free ultrasound image, $\theta(x, y)$ is additive white Gaussian noise (AWGN) with zero-mean and variance $\sigma^2$, and $r$ is associated with ultrasonic equipment. A large number of studies have shown that $r = 0.5$ is the best value that can be used to simulate speckle noise in ultrasonic images (*Yu et al., 2018*; *Lan & Zhang, 2020*).

### Denoising network

The architecture of denoising network is shown in Fig. 2A. The denoising network is based on U-Net, which consists of a contracting path and an expanding path. The expanding function of the decoder module is to gradually restore the spatial and boundary information. The contracting function of the encoder module is to gradually reduce the spatial dimensions and capture high-level feature information (*Zhang et al., 2020*). Nevertheless, these successive convolutions and pooling layers cause the loss of spatial information. Additionally, the problem of vanishing gradient is a key point that hinders the networks from training as the networks deepen. Some densely connected methods capture more information and avoid the appearance of vanishing-gradient problem (*Huang et al., 2017*; *Zhang et al., 2018*; *Park, Yu & Jeong, 2019*). Inspired by these methods,

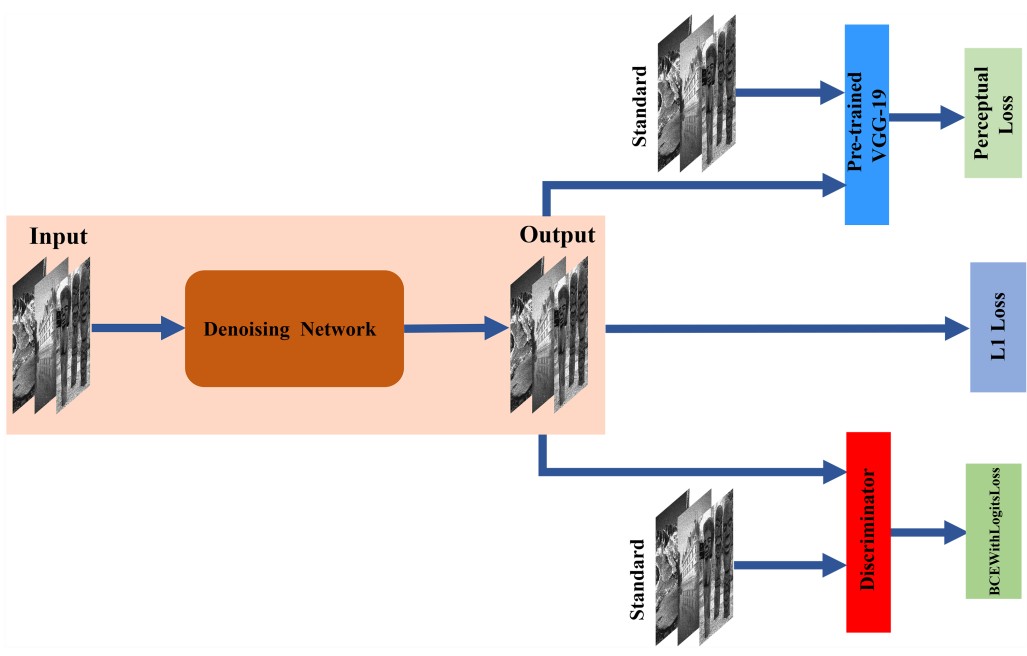

**Figure 1 Architecture of the proposed network.** The denoising network is tasked with translating input images to the target domain through encoder–decoder networks. The discriminator is trained to distinguish between standard and denoising images. The pre-trained VGG-19 is used to acquire more features as perceptual loss.

we applied two residual dense connectivity blocks (RDCBs) to each module of the encoder and decoder modules. The architecture of RDCBs is shown in Fig. 2B. The RDCBs is composed of three convolutional layers followed by BN and ReLU. Each module applies the previous feature map through dense connectivity. We adopt dense connectivity and local residual learning to improve the information flow so that the proposed algorithm can avoid the vanishing gradient problem and accurately remove speckle noise. Meanwhile, RDCBs can capture more features to improve denoising performances.

The network architecture of the encoder module is shown in Fig. 2C. The encoder module is composed of two RDCBs, a downsampling module and a convolution module. The downsampling module is a 2×2 max-pooling layer. The convolutional module is a 1×1 convolution layer followed by BN and ReLU. The feature map is fed into two RDCBs to preserve more feature information and avoid vanishing gradient. Subsequently, the feature map is fed into 2×2 max-pooling layers decreasing the size of feature map. Finally, the feature map is fed into a 1×1 convolution layer followed by BN and ReLU. The size of the output feature maps of the encoder module is half the size of the input feature maps.

The architecture of the decoder module is shown in Fig. 2D. It is the inverse process of the encoder module. It consists of three modules: two RDCBs, a 1×1 convolution layer followed by BN and ReLU and a subpixel interpolation layer. We use a 1×1 convolution layer to refine the feature maps. Compared with the 2×2 deconvolution layer, subpixel interpolation can expand the feature maps size more accurately and efficiently. Therefore,

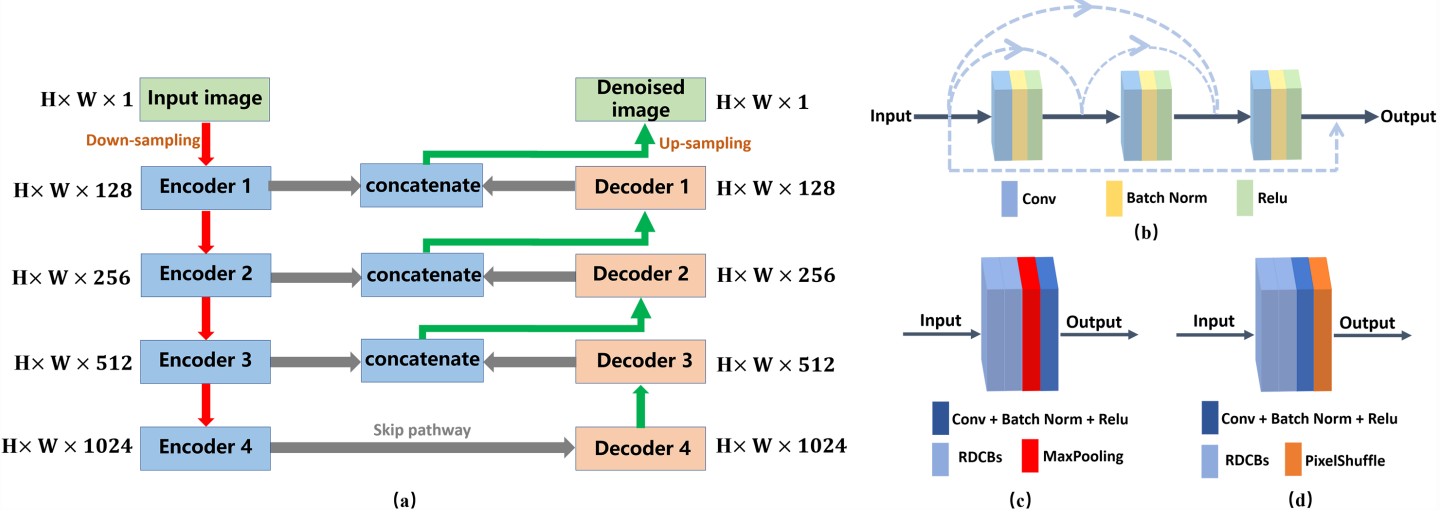

**Figure 2  Architecture of the denoising network.** (A) Architecture of the denoising network. H ×W ×C specify the output dimensions of each component (*C* = 1, 128, 256, 512, 1024). (B) Architecture of the RDCBs. (C) The architecture of the encoder module. (D) The architecture of the decoder module. Conv denotes a 3 ×3 convolution layer.

the size of the output feature map of the upsampling block is twice the size of the input feature map, and the number of channels of the input feature map is one second.

## Discriminator

The discriminator is trained to distinguish the difference between the denoising image and the standard image, where the denoising attempts to fool the discriminator. It uses a set of convolutional layers to build a discriminative network. It consists of an input convolutional layer and nine convolutional layers followed by BN and ReLU. The output channels of consecutive convolutional layers are 64, 128, 256, 512 and 1. Therefore, when the input image is passed through each convolution block, the spatial dimension is decreased by a factor of two. The architecture of the discriminator network framework for ultrasound image denoising is shown in Fig. 3.

## Loss function

Traditionally, learning-based image restoration uses the per-pixel loss between the restored image and ground truth as the optimization target, and excellent quantitative scores can be obtained. Nevertheless, in recent studies, relying only on low-level pixels to minimize pixelwise errors has proven that it can lead to the loss of details and smooth the results (*Johnson, Alahi & Li, 2016*). In this paper, we use a weighted sum of the loss function. It consists of the denoising loss, the perceptual loss of the feature extractor and the discriminator loss.

The denoising network loss is the L1 loss function, which minimizes the pixelwise differences between the standard image and the denoising image. The L1 loss is used and

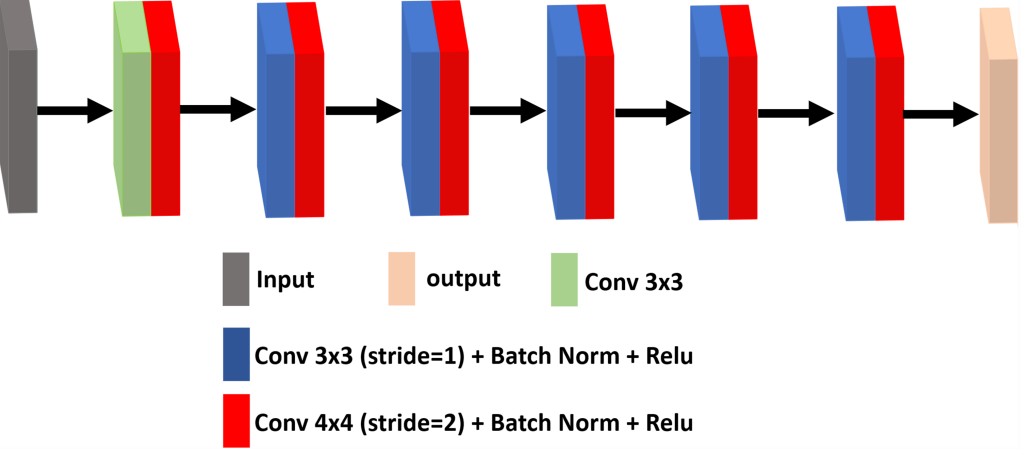

**Figure 3  The architecture of the discriminator network.**

calculated as follows:

$$L1 = \sum_{i=1}^{n} |x - y| \qquad (2)$$

where $x$ is the denoising image and $y$ is the corresponding ground truth.

Recent studies have shown that the target image and the output image have similar feature representations, not just every low-level pixel that matches them *Johnson, Alahi & Li (2016)*. The critical point is that the pretrained convolutional neural model used for image semantic segmentation or classification has learned to encode image features, and these features can be directly used for perceptual loss.

To preserve image details more effectively in removing noise, we use perceptual loss as one of the loss functions, which is calculated by:

$$L_{per} = \left\| \theta\left(y_{true}\right) - \theta\left(y_{out}\right) \right\|_2^2 \qquad (3)$$

where $\theta$ represents the feature extraction operator of the pretrained network. The convolution neural network pre-trained for image classification which has already learned to capture features. These features can be used as perceptual loss. In our proposed method, we adopt the output before the first pooling layer from the pretrained VGG-19 network to extract features as perceptual loss (*Gong et al., 2018*). To the discriminator network, we use BCEWithLogitsLoss to discern the output image quality from the denoising network and the standard image. Then, we obtain the weighted joint loss function, which consists of L1 loss($L1$), perceptual loss ($Lper$) and BCEWithLogitsLoss ($L_{BCE}$). $\lambda_1$, $\lambda_2$, $\lambda_3$ are scalar weights for $L_{loss}$.

$$L_{loss} = \lambda_1 L_1 + \lambda_2 L_{per} + \lambda_3 L_{BCE}. \qquad (4)$$

$$\lambda_1 = 1, \lambda_2 = 0.1, \lambda_3 = 1.$$

## Training and testing details

To train our network, we use the Berkeley segmentation dataset (BSD400) composed of 400 images of size $180 \times 180$ for training (*Martin et al., 2001*; *Zhang et al., 2017*; *Chen & Pock, 2016*). Then, according to Eq. (1), speckle noise is added to the datasets and the noisy images are generated. For training data that have three noise levels, we train the model for speckle denoising with noise levels $\sigma =15$, 25 and 50 independently. We set the patch sizes to $40 \times 40$ to train our model. To avoid overfitting, we apply data augmentation by randomly rotating and flipping. The initial learning rate is set to $1e-4$ and halved every 2000 epochs. We use Adam optimizer and a batch size of 32 during training.

For the test images, we adopt Berkeley segmentation (BSD68) (*Martin et al., 2001*; *Roth & Black, 2009*) datasets for grey synthetic noisy images, which include 68 natural images, $321 \times 481$ or $481 \times 321$ in size. To further verify the practicality of the proposed GAN-RW method, we also illustrate the results of our method as well as eight existing denoising methods for ultrasound images from the Kaggle Challenge (*Rebetez, 2016*), the Grand Challenge (*Thomas et al., 2018*) and lymph node datasets (*Zhang, Wang & Shi, 2009*). We applied PyTorch (version 1.7.0) as the framework to implement our network. Training takes place on a workstation equipped with an NVIDIA 2080Ti graphic card with 11 GB of memory.

There is a phenomenon that deep learning-based networks with the same training data and seed points will get different results. Therefore, we repeated each training three times with the same parameters and seeds, and then used the results of three experiments on test datasets to obtain the mean value and standard deviation.

## Evaluation metrics

In order to test the performance of the proposed method, the peak signal-to-noise ratio (PSNR) (*Chan & Whiteman, 1983*) and the structural similarity (SSIM) (*Wang et al., 2004*) are used to verify quantitative metrics. Meanwhile, the denoising results are used to show the visual quality of denoising images. If the denoising method has higher the PSNR and SSIM results on the test datasets, the denoising network shows better performance. In addition, to clarify the visual effect on the denoised images, we zoom in on the area of the denoising image for display. If the magnified area is clearer, it shows that the denoising method is more effective than others.

## RESULTS

To demonstrate the superiority of our proposed method in despeckling effect, we compared our proposed network with deep learning-based methods and traditional denoising algorithms. The methods for these comparisons were as follows: BM3D (*Dabov et al., 2007*), DnCNN (*Zhang et al., 2017*), DnCNN_Enhanced (*Jifara et al., 2019*), BRDNet (*Tian, Xu & Zuo, 2020*), DHDN (*Park, Yu & Jeong, 2019*), CBDNet (*Guo et al., 2019*), MuNet (*Lee et al., 2020*) and EDNet (*Couturier, Perrot & Salomon, 2018*). Two performance metrics are used, namely, PSNR and SSIM, which are expressed in terms of average value and standard deviation. Statistical analysis was performed with SPSS statistics software (version 26.0; IBM Inc., Armonk, NY, USA). All deep-learning based methods were trained three

**Table 1** **The mean and standard deviation of the PSNR (dB) and SSIM values of different methods for denoising BSD68 gray images.** The best result is highlighted with bold.

| Method | BSD68 | | | | | |
| --- | --- | --- | --- | --- | --- | --- |
| | $\sigma = 15$ | | $\sigma = 25$ | | $\sigma = 50$ | |
| | PSNR | SSIM | PSNR | SSIM | PSNR | SSIM |
| Noisy | 32.06 | 0.8377 | 27.71 | 0.6984 | 21.86 | 0.4599 |
| BM3D | 33.30 ± 0.2178 | 0.9045 ± 0.0045 | 30.62 ± 0.1747 | 0.8512 ± 0.0055 | 27.43 ± 0.1737 | 0.7683 ± 0.0049 |
| DnCNN | 33.86 ± 0.0043 | 0.9332 ± 0.0001 | 30.98 ± 0.0091 | 0.8862 ± 0.0002 | 27.24 ± 0.0124 | 0.7907 ± 0.0005 |
| DnCNN- Enhanced | 33.87 ± 0.0036 | 0.9332 ± 0.0004 | 31.00 ± 0.0057 | 0.8862 ± 0.0006 | 27.22 ± 0.0066 | 0.7907 ± 0.0007 |
| BRDNet | 33.79 ± 0.0202 | 0.9319 ± 0.0006 | 30.95 ± 0.0283 | 0.8843 ± 0.0015 | 27.20 ± 0.0140 | 0.7888 ± 0.0018 |
| DHDN | 35.18 ± 0.0203 | 0.9393 ± 0.0003 | 32.03 ± 0.0388 | 0.8938 ± 0.0013 | 27.62 ± 0.0374 | 0.8035 ± 0.0009 |
| CBDNet | 33.83 ± 0.0206 | 0.9334 ± 0.0004 | 31.01 ± 0.0145 | 0.8875 ± 0.0006 | 27.24 ± 0.0161 | 0.7926 ± 0.0015 |
| MuNet | 34.67 ± 0.1099 | 0.9296 ± 0.0024 | 31.51 ± 0.0956 | 0.8780 ± 0.0022 | 27.53 ± 0.1049 | 0.7929 ± 0.0035 |
| EDNet | 34.07 ± 0.3938 | 0.9277 ± 0.0039 | 31.42 ± 0.1916 | 0.8799 ± 0.0049 | 27.53 ± 0.0261 | 0.7993 ± 0.0022 |
| GAN-RW-WD | 35.13 ± 0.0334 | 0.9389 ± 0.0003 | 31.98 ± 0.0378 | 0.8919 ± 0.0009 | 27.63 ± 0.0537 | 0.8015 ± 0.0028 |
| GAN_RW | **35.28 ± 0.0193** | **0.9404 ± 0.0004** | **32.15 ± 0.0243** | **0.8969 ± 0.0010** | **27.74 ± 0.0151** | **0.8064 ± 0.0003** |

times and BM3D used three different parameters to obtain the average value and standard deviation. Experiments were performed on the BSD68 and ultrasound images.

## The BSD68

Table 1 shows the mean, standard deviation of the proposed methods and the compared methods for the BSD68 test datasets. In Table 1, the best result is highlighted in bold. When the noise level was 15, the average PSNR and SSIM of our proposed method improved by 1.21dB and 0.0113, which were better than those of the compared method. The average performance of the GAN-RW increased by approximately 3.58% and 1.23% for PSNR and SSIM, respectively. When the noise level was 25, the average PSNR and SSIM of this method improved by 0.96dB and 0.0160, and increased by approximately 3.08% and 1.84% for PSNR and SSIM, respectively. When the noise level was 50, the average PSNR and SSIM of this method improved by 0.36dB and 0.0156 and increased by approximately 1.32% and 1.98% for PSNR and SSIM, respectively. As shown in Table 1, the proposed method is superior to the traditional methods for three noise levels.

To compare subjective performance, we compared the denoising images for different methods. Figures 4, 5 and 6 show the grey scale denoising image of the proposed methods and the compared method at different noise levels. To easily observe the performance of GAN-RW and other methods, we zoomed in on an area from denoising images obtained using the compared methods. In Fig. 4, the proposed method accurately restored the pattern, while the compared methods achieved blurred denoising image. As shown in Fig. 5, the compared methods failed to exactly restore the windows or achieved blurred denoising image. However, the proposed method restored the windows accurately. Similarly, unlike the compared methods, the details of the zebra stripes could not be restored. The proposed method restored the details in Fig. 6. As shown in these images under different noise levels,

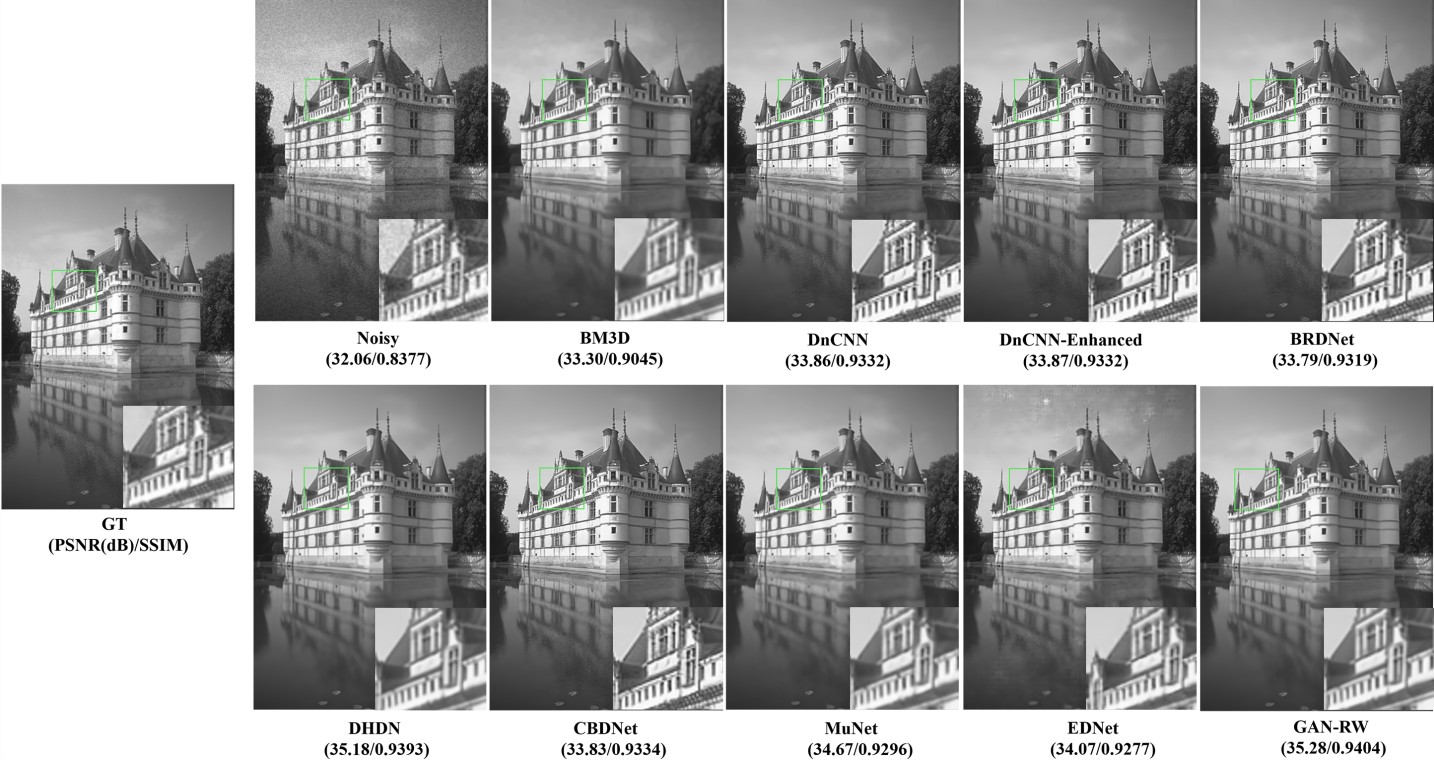

**Figure 4** Speckle denoising results of the compared methods and the proposed method on noise level $\sigma = 15$.

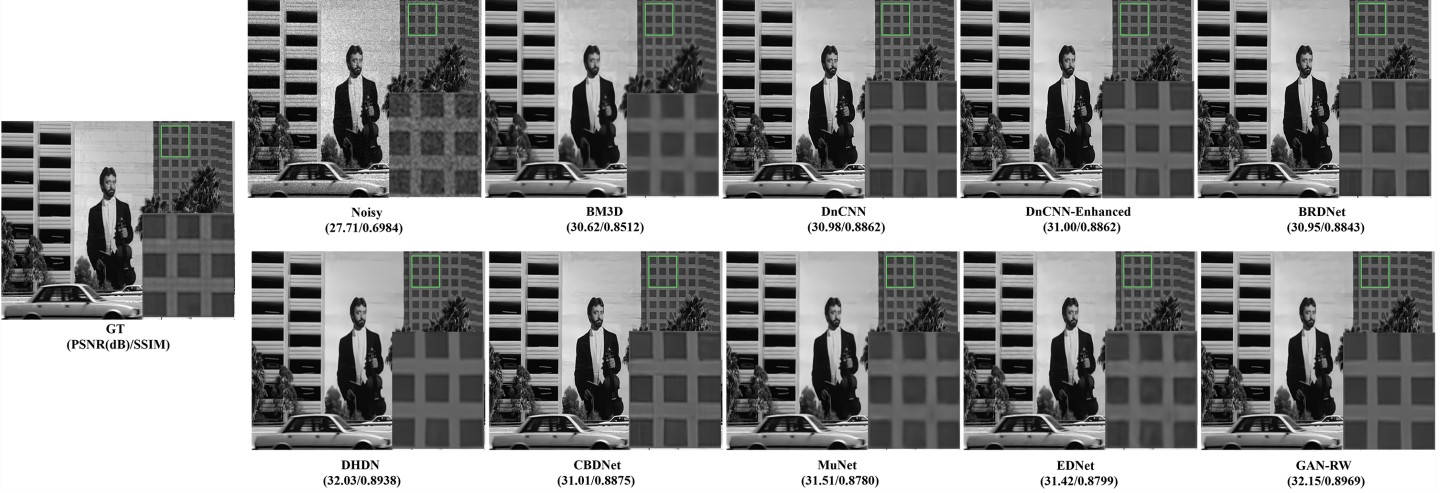

**Figure 5** Speckle denoising results of the compared methods and the proposed method on noise level $\sigma = 25$.

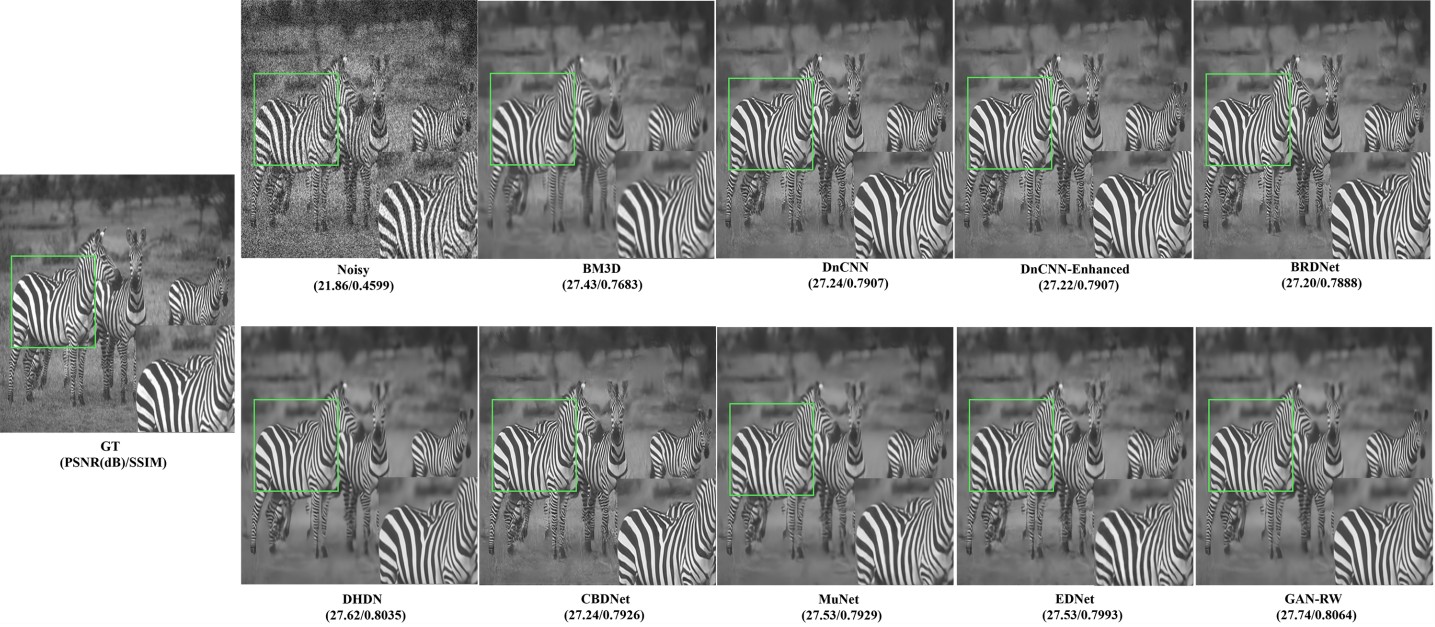

**Figure 6** Speckle denoising results of the compared methods and the proposed method on noise level $\sigma = 50$.

the traditional methods produced blurred results and could not restore the details of the patterns, while the proposed method accurately restored the patterns.

## Ultrasound images

We used the ultrasound images of lymph nodes, the foetal head and the brachial plexus with a noise level of 25 to verify the practicality of the proposed GAN-RW. To observe the performance of GAN-RW and other eight existing algorithms, we marked the fine details with red box in the figure. Figure 7 shows the despeckling images of different methods on the lymph node ultrasound image. Compared methods either failed to removed noise effectively or produced blurry and artifact results. Obviously, the results showed that the proposed method effectively removed speckle noise while better retaining image details and improving the subjective visual effect.

In addition, other foetal ultrasound images were applied to visually compare the despeckling performance of all evaluated methods. In Fig. 8, it is easy to observe that our proposed algorithm produced a smoother outline and retained the image details better than the other methods.

In the end, we compare the different methods on the brachial plexus ultrasound images. In Fig. 9, the proposed GAN-RW can smoother background regions and preserve image hierarchy structure information better than the other methods.

## Ablation study

To justify the effectiveness of the RDCBs, we conducted the following experiments on BSD68. In a section of denoising network, RDCBs is composed of three convolutional

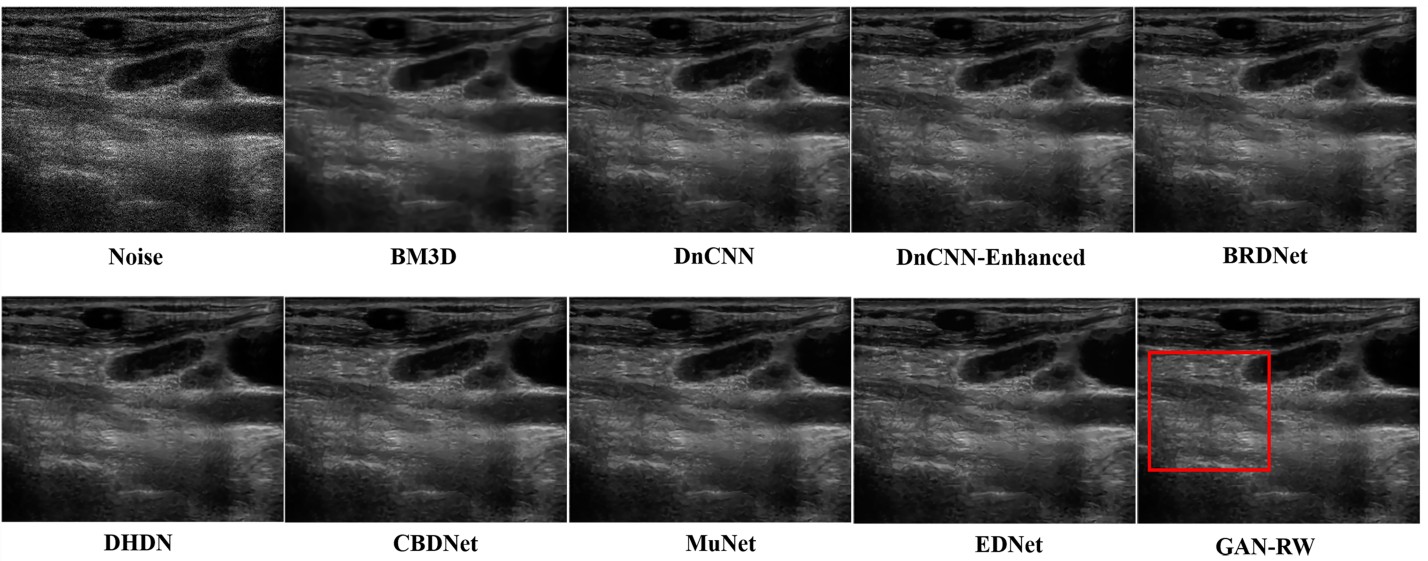

**Figure 7 Speckle denoising results of the compared methods and the proposed method on the real ultrasound images of Lymph nodes.**

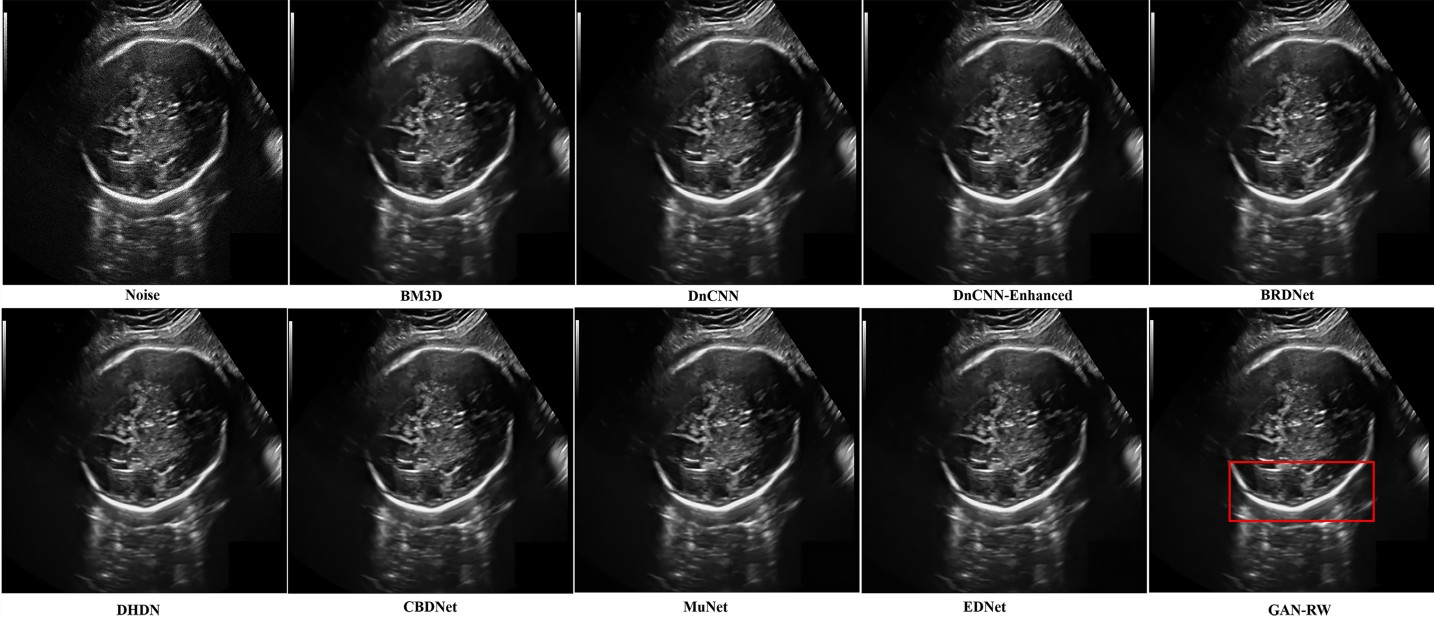

**Figure 8 Speckle denoising results of the compared methods and the proposed method on the real ultrasound images of foetal head.**

layers followed by BN and ReLU and each module applied the previous feature map through dense connectivity. We used two successive convolutional layers followed by BN and ReLU without dense connectivity (GAN-RW-WD) to replace two RDCBs. The experimental results compared with GAN-RW are shown in Table 1. When the noise level was 15, RDCBs can enhance the average PSNR by approximately 0.44% and the average

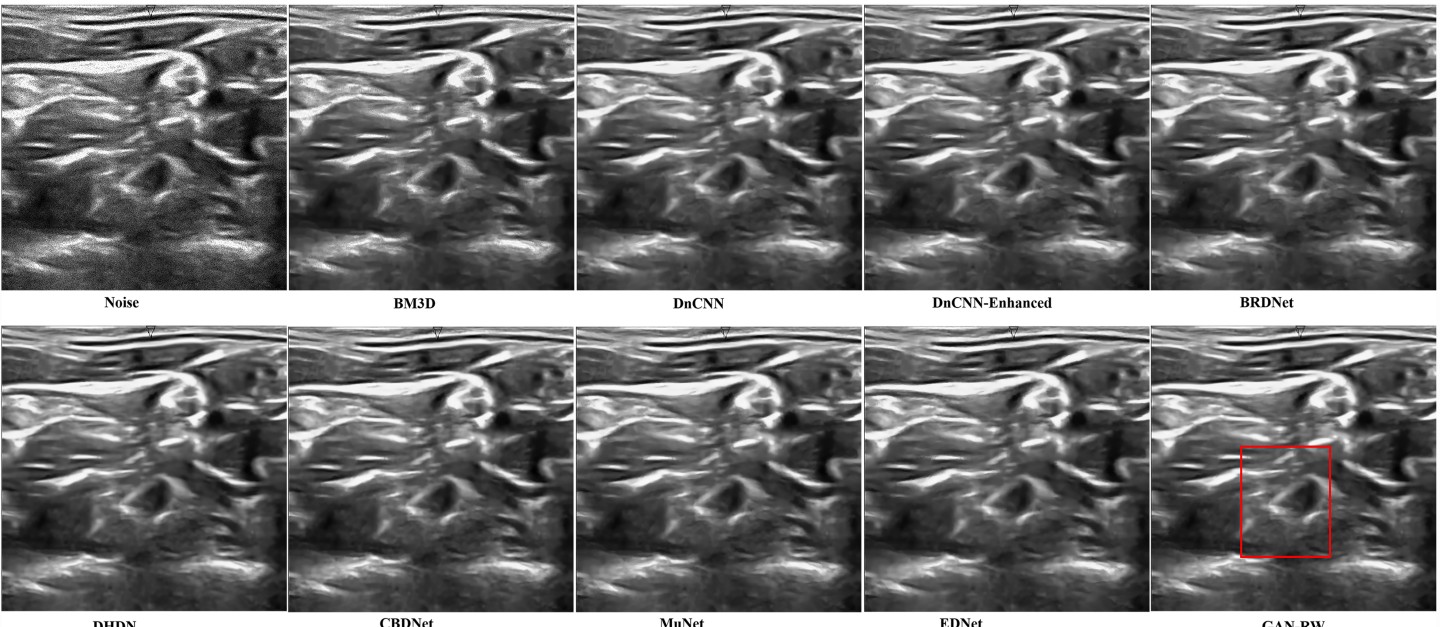

**Figure 9** Speckle denoising results of the compared methods and the proposed method on the real ultrasound images of Brachia Plexus.

SSIM by approximately 0.16% for the BSD68, respectively. When the noise level was 25, RDCBs can enhance the average PSNR by approximately 0.53% and the average SSIM by approximately 0.57% for the BSD68, respectively. When the noise level was 50, RDCBs can enhance the average PSNR by approximately 0.37% and the average SSIM by approximately 0.61% for the BSD68, respectively.

## Statistical analysis

Statistical analysis is necessary to verify the superiority of the proposed method. Due to the PSNR and SSIM values were not Gaussian distribution, we used the nonparametric Friedman test (*Friedman, 1937*) to assess the performance of different denoising algorithms. The mean rank and p-Value of PSNR and SSIM of all algorithms are shown in Table 2. Usually, a p-value of less than 0.05 is deemed the significant difference. The mean rank presents the performance of different algorithms, and the higher value of mean rank has the better performance. It can be seen from Table 2 that GAN-RW has a significant improvement over other algorithm.

## DISCUSSION

In this paper, we proposed a generative adversarial network for ultrasound image despeckling. The GAN-RW is based on U-Net with residual dense connectivity, BN and a joint loss function to remove speckle noise. We used natural images and ultrasound images to verify our method.

For the BSD68 test datasets, when the noise level was 15, our method achieved 35.28dB and 0.9404 for PSNR and SSIM. Compared with the original noise image, the average

**Table 2  The mean rank (Friedman test) of the PSNR (dB) and SSIM values of the different methods for denoising BSD68 gray images.**

| Methods | $\sigma = 15$ | | | | $\sigma = 25$ | | | | $\sigma = 50$ | | | |
|---|---|---|---|---|---|---|---|---|---|---|---|---|
| | PSNR | | SSIM | | PSNR | | SSIM | | PSNR | | SSIM | |
| | Mean Rank | p-Value | Mean Rank | p-Value | Mean Rank | p-Value | Mean Rank | p-Value | Mean Rank | p-Value | Mean Rank | p-Value |
| BM3D | 1.90 | | 1.47 | | 2.12 | | 1.69 | | 4.96 | | 2.39 | |
| DnCNN | 4.32 | | 5.16 | | 3.56 | | 4.98 | | 3.91 | | 4.26 | |
| DnCNN-Enhanced | 4.34 | | 5.32 | | 4.03 | | 5.21 | | 3.10 | | 4.00 | |
| BRDNet | 2.13 | 0.000 | 3.26 | 0.000 | 2.29 | 0.000 | 3.25 | 0.000 | 2.21 | 0.000 | 2.77 | 0.000 |
| DHDN | 7.97 | | 7.65 | | 7.81 | | 7.37 | | 7.46 | | 7.09 | |
| CBDNet | 3.75 | | 6.07 | | 4.26 | | 6.38 | | 4.22 | | 5.53 | |
| MuNet | 6.51 | | 3.51 | | 6.13 | | 2.96 | | 5.60 | | 4.66 | |
| EDNet | 5.18 | | 3.87 | | 6.07 | | 4.63 | | 6.06 | | 5.99 | |
| GAN-RW | **8.90** | | **8.67** | | **8.72** | | **8.53** | | **7.49** | | **8.31** | |

values of the GAN-RW increased by approximately 10.05% and 12.26% for PSNR and SSIM, respectively. When the noise level was 25, our method achieved 32.15dB and 0.8969 for PSNR and SSIM, respectively. Compared with the original noise image, the average performance of the GAN-RW increased by approximately 16.01% and 28.43% for PSNR and SSIM. When the noise level was 50, our method achieved 27.74dB and 0.8064 for PSNR and SSIM, respectively. Compared with the original noise image, the average performance of the GAN-RW increased by approximately 26.88% and 75.35% for PSNR and SSIM, respectively. In Fig. 10, boxplots show the comparison of PSNR and SSIM under different noise levels for BSD68. In the end, we used the ultrasound images of lymph nodes, the brachial plexus and the foetal head to verify the practicality of the proposed GAN-RW. In contrast, GAN-RW can effectively eliminate speckle noise while retaining image details better and improving the visual effect.

## CONCLUSIONS

In conclusion, we developed and verified a new ultrasound image despeckling method. GAN-RW is based on U-Net and uses residual dense connectivity, BN and joint loss functions to remove speckle noise. Compared with BM3D, DnCNN, DnCNN-Enhanced, BRDNet, DHDN, CBDNet, MuNet, EDNet and GAN-RW achieves better despeckling performance on three fixed noise levels of BSD68. We also effectively verified the proposed method on ultrasound images of lymph nodes, the brachial plexus and the foetal head.

## ACKNOWLEDGEMENTS

We would like to thank Bo Li and Yang Zhao for their constructive discussion during the manuscript revision.

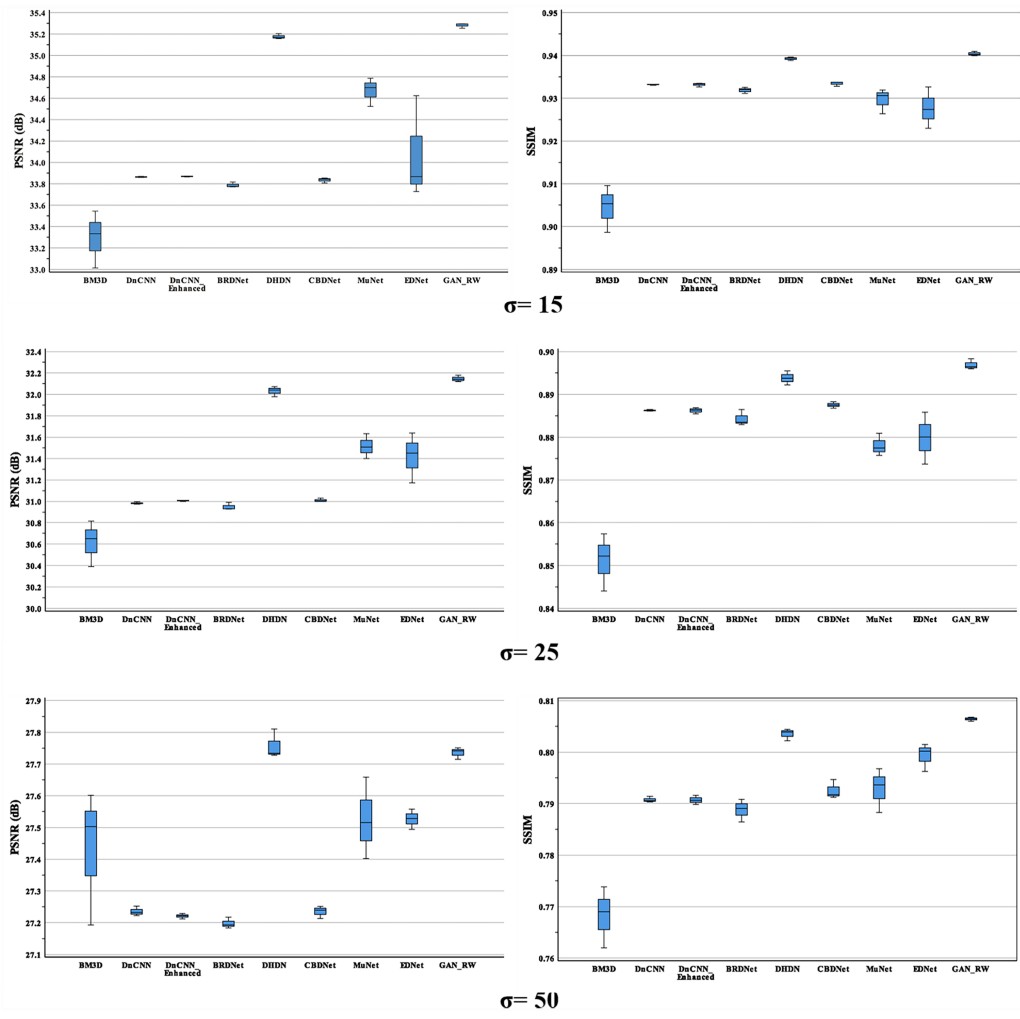

**Figure 10** Boxplots of average PSNR (dB) and SSIM results of compared methods and proposed method for BSD68.

## Funding

The research work was supported by the grants from the Natural Science Foundation of China under Grants 62063034 and 61841112. The funders had no role in study design, data collection and analysis, decision to publish, or preparation of the manuscript.

## Grant Disclosures

The following grant information was disclosed by the authors:
The Natural Science Foundation of China: 62063034, 61841112.

## Competing Interests

The authors declare there are no competing interests.

## Author Contributions

- Lun Zhang conceived and designed the experiments, performed the experiments, analyzed the data, performed the computation work, prepared figures and/or tables, authored or reviewed drafts of the paper, and approved the final draft.
- Junhua Zhang conceived and designed the experiments, analyzed the data, authored or reviewed drafts of the paper, and approved the final draft.

## Data Availability

The datasets of the BSD400, BSD68 and lymph node are available at GitHub: https://github.com/smartboy110/denoising-datasets.

The datasets of the foetal head are available at Zenodo: Thomas L. A. van den Heuvel, Dagmar de Bruijn, Chris L. de Korte, & Bram van Ginneken (2018). Automated measurement of fetal head circumference using 2D ultrasound images [Data set]. Zenodo. https://doi.org/10.5281/zenodo.1327317.

The datasets of the brachial plexus are available at Kaggle: https://www.kaggle.com/c/ultrasound-nerve-segmentation.

Figure 5 and the BSD68 figures are available from the Berkeley Segmentation Dataset and Benchmark (https://www2.eecs.berkeley.edu/Research/Projects/CS/vision/bsds/).

## Supplemental Information

Supplemental information for this article can be found online at http://dx.doi.org/10.7717/peerj-cs.873#supplemental-information.

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
