# Peer review of "Ultrasound image denoising using generative adversarial networks with residual dense connectivity and weighted joint loss"

_PeerJ Computer Science, doi:10.7717/peerj-cs.873_

## Round 0.1 · original submission · Major Revisions

Two consistent reviews have been received. I agree with the comments that a major revision is needed before a further consideration.

·

Basic reporting

The authors of the article named "Ultrasound Image Denoising Using Generative Adversarial Networks with residual dense connectivity and weighted joint loss" propose a novel method based on the GAN network for denoising of speckle noise in images. They train and evaluate their method using the SNR and SSIM metrics. They compare the results with the other state-of-the-art methods in image denoising and prove their GAN outperforms other methods.

The language is clear, consistent, and professional English. The literature overview is sufficient, however, there could be a few improvements and additional references. On line 78 you write "(...) methods (...), which 'are' divided into two categories." If it is common to divide these methods in these categories then use a reference, if not, write "could be" instead of "are". There should be a citation for the SSIM. In line 252 you should cite all of the named models.

The article is well structured. Figures could be improved. Figures 9 and 12 are mentioned in the article as "Figure", others are mentioned as "Fig.". The authors should fix that. In figure 12, the authors should add what dataset does the bar plot refers to.

Experimental design

Research is well defined and structured. The results are interesting and well presented. However, in Table I. authors should add the average value and standard deviation. Models should be trained multiple times to avoid accidental results.

The authors should mention what does the skip pathway refers to, concatenation or addition? Authors could try to squeeze the whole model in one or two figures for better clarity and generally improve Figure 1.

Methods to which the GAN is compared could be better described.

Validity of the findings

The article proposes original work, it is impactful and has some novelty. All data has been provided. Conclusions are well stated and limited to supporting results.

Additional comments

The general impression of the article is good, I hope the authors will submit the review. The headline of the article is based on the ultrasound images. However, the method has only been visually evaluated on the ultrasound dataset, all analysis has been done on some other publicly available datasets.

Reviewer 2 ·

Basic reporting

Literature needs improvement. Writing style is OK, but Figures are poorly drawn, requires directional arrows to illustrate the dataflow

Experimental design

OK

Validity of the findings

Its OK. Please refer to my final comments related to the noise levels

Additional comments

Major Concerns :

P8 Lin-79- The author classify the Wavelet method as a Frequency domain techniques, which is absurd. Wavelets provide both time and frequency. This statement/ and preceding ones should be modified accordingly

P9, Lin 106. What does modified UNET mean. Provide reasonable information about the modifications. Also there are many UNET variations which need to be mentioned in literature - RDAU-NET, RDA-UNET-WGAN, ADID UNET, UNET++, VGGUNET, Ens4B-UNet etc. Though these models are NOT directly related to denoising, it is worth mentioning these models atleast in the literature

P10 Lin 138 - The Speckle Noise Model requires more explanation. The current information is minimalistic.

P11. Lin 155 - Please change loss of image information to loss of spatial information

P11 Lin 158 - Use of Residual blocks and Dense nets are known to avoid vanishing gradient problem, as stated in RDAUNET and ADID UNETS, hence what additional improvement does RDCB provide

P14, Lin 212. I am confused as to why the author is explaining VGG 19 model ?. I feel the VGG part can be removed or rewritten in way that is related to the proposed architecture


P15, Lin 245, Is Visual effect ( the performance metric as explained by the author ) is used elsewhere in the literature ? If so please cite those papers .

Figure 1 requires directions (arrows) to illustrate the flow the data. At the moment, its really confusing to the readers to understand how the entire model works

Figure 12. Why does the noise levels vary between the same sigma value for PSNR and SSIM. Does noise level expressed in db should be same (eq 32db in PSNR and 0.8377 in SSIM) ?

In general the paper requires more additional literature in regards to UNETs and GANs. the authors have to clearly mention the term " Modified " that is used in their model

---

## Round 0.2 · Minor Revisions

A minor revision is needed before further processing. I look forward to receiving your revised version.

·

Basic reporting

References are improved. Professional english is used.

Experimental design

Research question is now well defined. Methods description is improved as well as figures. However, authors do not state how many times were experiments done to prove the robustnes of the methods, at least I do not see it.

Validity of the findings

Conclusions are well stated, and all underlying data has been provided.

Additional comments

If these small modifications are made the article is suitable for publishing, in my opinion.

---

## Round 0.3 · accepted · Accept

I recommend publication of the paper.